# Prognostic Relevance of Circulating Tumor Cells and Circulating Cell-Free DNA Association in Metastatic Non-Small Cell Lung Cancer Treated with Nivolumab

**DOI:** 10.3390/jcm8071011

**Published:** 2019-07-10

**Authors:** Angela Alama, Simona Coco, Carlo Genova, Giovanni Rossi, Vincenzo Fontana, Marco Tagliamento, Maria Giovanna Dal Bello, Alessandra Rosa, Simona Boccardo, Erika Rijavec, Federica Biello, Luca Longo, Zita Cavalieri, Cristina Bruzzo, Francesco Grossi

**Affiliations:** 1Lung Cancer Unit, Division of Medical Oncology II, IRCCS Ospedale Policlinico San Martino, Largo R. Benzi 10, 16132 Genova, Italy; 2Clinical Epidemiology Unit, IRCCS Ospedale Policlinico San Martino, Largo R. Benzi 10, 16132 Genova, Italy; 3Medical Oncology, Fondazione IRCCS Ca’ Granda Ospedale Maggiore Policlinico, Via F. Sforza, 28, 20122 Milan, Italy; 4AOU Maggiore della Carità, Corso Mazzini 18, 28100 Novara, Italy

**Keywords:** liquid biopsy, lung cancer, circulating tumor cells, cell-free DNA, immune checkpoint inhibitors, nivolumab, prognosis

## Abstract

The treatment of advanced non-small cell lung cancer (NSCLC) has been revolutionized by immune checkpoint inhibitors (ICIs). The identification of prognostic and predictive factors in ICIs-treated patients is presently challenging. Circulating tumor cells (CTCs) and cell-free DNA (cfDNA) were evaluated in 89 previously treated NSCLC patients receiving nivolumab. Blood samples were collected before therapy and at the first and second radiological response assessments. CTCs were isolated by a filtration-based method. cfDNA was extracted from plasma and estimated by quantitative PCR. Patients with baseline CTC number and cfDNA below their median values (2 and 836.5 ng from 3 mL of blood and plasma, respectively) survived significantly longer than those with higher values (*p* = 0.05 and *p* = 0.04, respectively). The two biomarkers were then used separately and jointly as time-dependent covariates in a regression model confirming their prognostic role. Additionally, a four-fold risk of death for the subgroup presenting both circulating biomarkers above the median values was observed (*p* < 0.001). No significant differences were found between circulating biomarkers and best response. However, progressing patients with concomitant lower CTCs and cfDNA performed clinically well (*p* = 0.007), suggesting that jointed CTCs and cfDNA might help discriminate a low-risk population which might benefit from continuing ICIs beyond progression.

## 1. Introduction

The introduction of immune checkpoint inhibitors (ICIs) in the treatment of advanced non-small cell lung cancer (NSCLC) has radically changed the management of this tumor [1]. Inhibitors of programmed death 1 (PD-1) on activated T lymphocytes and its ligand (PD-L1), expressed on cancer cells, have significantly improved the overall survival (OS) of previously treated [2,3,4,5] and previously untreated metastatic NSCLC when given alone or in combination with chemotherapy [6,7,8]. Nivolumab is an ICI that blocks the interaction between PD-1 and PD-L1 enhancing the immune T-cell response. The recent advancement in the management of NSCLC with ICIs requires the identification of reliable biomarkers which are able to select patients who can potentially obtain greater clinical benefit [9,10]. It has been observed that the evaluation of circulating biomarkers by liquid biopsy may provide useful prognostic and predictive information in advanced NSCLC, where tissue biopsy is particularly challenging [11,12]. Liquid biopsy is a blood test that can detect circulating tumor cells (CTCs) and tumor-derived nucleic acids such as cell-free DNA (cfDNA) and microRNAs, spread from the tumor into the bloodstream of cancer patients [13,14,15,16]. Several studies in advanced NSCLC concur that baseline CTC enumeration, as well as quantification of cfDNA, may significantly correlate with survival in untreated patients and also predict response to chemotherapy [17,18,19,20,21] or targeted therapies [22,23].

At present, the potential value of liquid biopsy within immunotherapy trials in advanced NSCLC is poorly defined, and only a few studies have addressed this issue. Emerging data suggest that changes in CTC or cfDNA blood levels during immunotherapy seem to be associated to clinical outcome [24,25,26,27,28], although no study has concomitantly explored their prognostic implication. In this study, we investigated the association of circulating biomarkers with survival and therapy response in 89 previously treated metastatic NSCLC patients receiving nivolumab, with the aim of defining the prognostic role of CTCs and cfDNA, separately and in conjunction.

## 2. Patients and Methods

### 2.1. Patients

Eighty-nine consecutive patients, with previously treated advanced NSCLC, were enrolled from May 2015 to April 2017 at the IRCCS Ospedale Policlinico San Martino (Genoa, Italy) within the Italian Nivolumab Expanded Access Program (EAP). The EAP was designed to allow Italian NSCLC patients to receive nivolumab in the timespan between its registration and its actual availability in our Country (NCT02475382). Eligible patients received nivolumab 3 mg/kg every 14 days. Nivolumab was administered until onset of unacceptable toxicities, patient refusal, radiologically confirmed disease progression, death, or upon reaching 96 weeks from the start of treatment. The enrolled patients were asked to give an additional informed consent for translational research, which was approved by our Institutional Ethics Committee (registry number: P.R. 191REG2015).

### 2.2. Blood Sample Collection

Blood samples (10–12 mL) were collected in ethylenediamine tetraacetic acid (EDTA)-containing tubes before the start of treatment with nivolumab (baseline) and after four (first radiological evaluation) and eight (second radiological evaluation) cycles of therapy. Radiologic assessment by computed tomography scan (CT-scan) was performed at every four cycles and response to nivolumab was evaluated according to the response evaluation criteria in solid tumors (RECIST *v*.1.1) as follows: complete response (CR), partial response (PR), stable disease (SD) and progressive disease (PD).

### 2.3. Circulating Tumor Cell Isolation and Enumeration

Circulating cells were isolated from 3 mL of whole peripheral blood by the filtration-based device ScreenCell CYTO (ScreenCell) according to manufacturer’s protocol. Circulating cells were isolated through a micro-porous membrane filter allowing only cells larger than the pores (7.5 ± 0.36 µm) to be retained on the membrane. The filter was then released on a slide, stained with haematoxylin-eosin (H&E) and observed under a light microscope. The isolated non-hematologic circulating cells with malignant features were defined as CTCs and morphologically identified and enumerated according to: nuclear size greater than or equal to 20 µm; high nuclear/cytoplasmic ratio (≥0.75); dense hyperchromatic nucleus; irregular nuclear membrane. Isolated cells with uncertain malignant phenotype were further assessed by immunofluorescence (Appendix A).

### 2.4. Circulating Free DNA Isolation and Quantification

Plasma was isolated, within at least 3 h from the blood withdrawal, by two consecutive steps of centrifugation at 1600 rpm for 15 min and then stored at −80 °C. cfDNA was extracted from 3 mL of plasma using the QIAamp Circulating Nucleic Acid Kit (Qiagen) according to the manufacturer’s protocol. The quantification of cfDNA was performed by absolute quantitative PCR (qPCR) method, using as reference the telomerase reverse transcriptase (*hTERT*) gene located on chromosome 5 (Thermo Fisher Scientific). The standard curve was obtained by a serial dilutions of a standard DNA (Promega) from 330,000 ng/mL to 3.3 ng/mL to assuming that 1 DNA copy number corresponds to 3.3 ng/mL of DNA. Each qPCR plate included positive and negative controls and cfDNA sample was run in duplicate and the concentration was calculated by interpolation of the mean of cycle threshold values with the standard curve.

### 2.5. Statistical Methods

The prognostic role of CTCs and cfDNA on OS was explored using the Kaplan–Meier method and differences in survival probabilities were statistically assessed through the log-rank test. The relationship of CTCs and cfDNA with OS, adjusted for potential imbalances in baseline clinical variables (age at start of nivolumab, gender, histotype, ECOG PS, smoking status, number of metastases, prior lines of treatment), was estimated through the Cox regression modeling. Hazard ratio (HR), and corresponding 95% confidence intervals (95% CI), was used as a measure of relative risk of dying during the follow-up period. Statistical significance of HR was assessed using the likelihood ratio test. For all statistical comparisons, a *p*-value ≤ 0.05 was considered statistically significant. All data analyses were performed using Stata (StataCorp. Stata Statistical Software, release 13.1 Statistical Software. College Station, TX, USA: StataCorp LP, 2013).

## 3. Results

### 3.1. Study Population

The baseline characteristics of the 89 enrolled patients are summarized in the Table 1. The median age was 67 years (range: 44.2–86.4) and most patients were males (70.8%). Histology was mainly represented by adenocarcinoma (70.8%) and a poor ECOG Performance Status (PS ≥ 1) accounted for 65.2% of cases. The majority of patients reported a smoking history (83%) and 63% had a number of metastases greater than three. Moreover, half of the patients received more than one line of treatment before nivolumab administration. The median OS was 8.1 months (range: 0.1–37.9).

### 3.2. Prognostic Significance of Circulating Biomarkers and Clinical Parameters

Blood withdrawals for liquid biopsy (CTCs and cfDNA) were executed prior to nivolumab treatment (baseline), at first and at second response assessments by CT-scans. CTC and cfDNA analyses were feasible in 89/89 patients at baseline, in 66/89 patients at the first evaluation and in 37 and 36 patients out of 89, respectively, at the second evaluation. CTCs at baseline were positive in 81/89 (91%) patients. Patients were dichotomized into two groups according to the median values of baseline CTC number and plasma cfDNA: 2 CTCs (range 0–21) and 836.5 ng (range 355.5–12,163.4) from 3 mL of blood and plasma, respectively. The role of circulating biomarkers in defining prognosis was investigated by the Kaplan–Meier method (Figure 1). Patients with lower values of CTCs and cfDNA survived significantly longer than those with higher levels (*p* = 0.05 and *p* = 0.04, respectively). In particular, median OS was 8.8 and 6.2 months in the CTC categories (≤2 vs. ≥2) and 9.4 and 5.1 months in the cfDNA categories (≤836.5 vs. >836.5).

Analyses of the association between CTCs and cfDNA and clinical variables did not provide substantial information; however, their potential confounding effect was considered in all multivariate analyses.

### 3.3. Time-Dependent Role of Circulating Biomarkers in the Overall Survival

To assess whether the repeated measurements of CTCs and cfDNA at first and second response assessments could provide additional insight for a clinical benefit, the two biomarkers were used separately and jointly as time-dependent covariates in a longitudinal Cox regression analysis. The Cox model was fitted to data firstly considering CTCs and cfDNA separately (i.e., the two biomarkers entered two distinct regression equations) and then jointly (i.e., both biomarkers were included in the same regression). After making an allowance for baseline clinical variables, statistically significant associations with OS probability were identified. Particularly, in the separate analysis patients with CTCs > 2 and cfDNA > 836.5 experienced a risk of dying which was almost twofold (CTCs: HR = 1.96, 95% CI = 1.12–3.42) and threefold (cfDNA: HR = 2.89, 95% CI = 1.58–5.29) compared with those below their medians. Similar results were also found when the circulating biomarkers were analyzed jointly (CTCs: HR = 1.72; 95% CI = 0.98–3.02; cfDNA: HR = 2.64; 95% CI = 1.44–4.83), as shown in Table 2.

These findings were also more straightforward when patients were categorized into four risk subgroups as follows: **1**. “low” (CTCs ≤ 2 and cfDNA ≤ 836.5); **2**. “low-intermediate” (CTCs > 2 and cfDNA ≤ 836.5); **3.** “high intermediate” (CTCs ≤ 2 and cfDNA > 836.5); **4.** “high” (CTCs > 2 and cfDNA > 836.5) as reported in Figure 2. More clearly, high-risk patients had a significantly worse clinical outcome and the risk of death was fourfold compared with those presenting lower biomarkers (*p* < 0.001).

### 3.4. Circulating Biomarkers and Response to Treatment

Among the 89 enrolled patients, 66 were suitable for response assessment. The disease control rate was 45% (14 PR and 16 SD), while 36 patients had PD as best response (BR). Among the 23 non-evaluable patients, 20 died before completing the first four cycles of therapy. The association of CTCs and cfDNA with BR was estimated according to their median cut-offs. No significant correlation was found between circulating biomarkers and BR (Table 3).

However, 23 out of 36 patients (64%) with PD had low biomarkers (CTC ≤ 2 and cfDNA ≤ 836.5). In order to estimate the “benefit-risk” of continuing nivolumab treatment in these PD patients, separate and joint analyses were evaluated by Cox modeling. Notably, the prognostic role of both biomarkers was retained by showing a twofold (HR = 1.90, 95% CI = 0.79–4.58) and fourfold (HR = 4.00, 95% CI = 1.54–10.40) increased risk of death with higher CTCs and cfDNA, respectively (Table 4).

Moreover, by considering the four risk subgroups emerged from the joint effect of cfDNA and CTCs (low, low-intermediate, high-intermediate and high), it is noteworthy that the death rate doubled as one moves from a lower to an upper category reaching a four-fold and eight-fold risk in the third (HR = 4.08, 95% CI = 1.55–10.7) and in the fourth (HR = 8.09, 95% CI = 2.06–31.7) category (Table 5).

## 4. Discussion

To date, the role of PD-L1 expression as biomarker in immunotherapy is still controversial [11,29,30], and the identification of reliable indicators that might assess prognosis and efficacy of immunotherapy is being strongly pursued [31,32].

Our study was focused on the prognostic role of two easy-to-measure biomarkers, which may be used as early predictors of disease progression and death. Previous studies have demonstrated the role of CTCs or cfDNA in defining prognoses and predicting response to treatment in advanced NSCLC [33,34,35,36], but few data in ICIs-treated patients have been reported to date. In the present study, we show that baseline CTCs and cfDNA, separately and in conjunction, are significantly associated to OS in NSCLC receiving nivolumab. The results obtained by the two markers seem to support their early role, when they are jointly considered as time-dependent covariates, even after adjusting for clinical variables. In this regard, we were able to identify a subgroup of patients, presenting both lower CTCs and cfDNA, who particularly benefited from the treatment, albeit in PD. Despite the small number of patients included in the different classes of risk of the overall population and PD patients, all four risk groups were fairly balanced and the HR and *p*-values emerged from the Cox analyses supported our hypothesis.

No significant association between circulating biomarkers and BR was shown, but in the subgroup of PD patients who were allowed to continue nivolumab, CTCs and cfDNA were still able to discriminate those benefiting from continuing ICI. To date, very few studies concerning CTCs and prognosis in NSCLC treated with ICIs have been described. Nicolazzo et al. examined the PD-L1 expression in CTCs of 24 stage IV NSCLC patients enrolled in EAP with nivolumab, as in our study. The authors showed that the presence of CTCs expressing PD-L1 on their surface (19 patients) was associated to poor outcome, while patients with PD-L1-negative CTCs achieved a clinical benefit after 6 months of treatment [24]. In our opinion these findings confirm the possible role of PD-L1 testing with nivolumab and support the prognostic role of CTCs in this setting. A more recent study by Guibert et al. also assessed the PD-L1 expression on CTCs in NSCLC receiving nivolumab after chemotherapy. In agreement with our data, patients with pre-treatment high CTC count experienced a worse outcome compared to those with a low CTC load. However, the presence of PD-L1-positive CTCs had no significant prognostic impact [25], confirming the ICI efficacy regardless of PD-L1 expression. Although the results on the role of PD-L1 were questionable, the prognostic value of CTC enumeration was confirmed and in line with our data. cfDNA has also been proposed as a prognostic and predictive biomarker in NSCLC [33,34,37,38]. In our study, the amount of cfDNA in plasma was assessed by qPCR using *hTERT* gene copy number that estimates the total cfDNA as surrogate of germinal and somatic portions. *hTERT* has already been used as reference gene in the cfDNA evaluation showing significant results [37,38,39], both in screening programs and chemotherapy regimens [36,38].

Currently, few studies have investigated the cfDNA as an indicator of OS in patients treated with ICIs. Similar to Giroux Leprieur et al., we observed that patients with lower cfDNA had a longer OS. In our analysis, significant results were obtained by considering the baseline median cfDNA, whereas Giroux Leprieur et al. found a significant correlation with cfDNA concentration at the first evaluation only [28]. A further divergence between the two studies was linked to the cfDNA cut-offs; however, this is not surprising, since the cut-off is often linked to the investigators’ choice and the methodology used. Recent studies have correlated the presence of cancer-associated somatic mutations in plasma (ctDNA), assessed by next-generation sequencing (NGS), with tumor response and survival [40,41]. In particular Iijima et al. reported that the presence of ctDNA was observed more frequently in patients with high tumor volumes, and that early changes in ctDNA levels within two weeks were associated to nivolumab efficacy [42]. Recently, Golberg et al. also investigated variations in ctDNA in relation to radiographic tumor size and OS in metastatic NSCLC receiving ICIs. The authors reported that a superior OS was significantly associated with a decrease in ctDNA greater than 50% of baseline [26]. More recently, a retrospective analysis with nivolumab in 45 previously treated advanced NSCLC patients demonstrated that increased cfDNA greater than 20% at the sixth week of nivolumab therapy was associated with worse OS [27]. The limited number of patients and the different methodologies of CTCs and cfDNA determinations described by the above studies make it particularly challenging to draw firm conclusions. To this end, few issues concerning our study with respect to the other investigations need to be considered. Our study examined the prognostic role of CTCs and cfDNA, either individually or in conjunction providing a more convincing indication of ICI benefit in a relative larger population of advanced NSCLC. Notably, the cohort of 89 patients in our analysis was enrolled by a single institution, and all the clinical and experimental data were exclusively collected among all the patients entered the study providing high uniformity in the sample processing and interpretation of results. Furthermore, as we previously demonstrated by a comprehensive range of 18 F-FDG PET/CT–derived parameters, the amount of cfDNA mainly reflects tumor metabolic activity (apoptosis, necrosis, phagocytosis, lysis), rather than tumor burden only [43]. Hence, the cfDNA analyzed in our clinical setting is more likely to be representative of the real proliferative and/or apoptotic status of the tumor that in turn mirrors the number of CTCs shed into the bloodstream. This assumption is confirmed by the highly significant association observed from the joint analysis of CTCs-cfDNA. Therefore, it can be hypothesized that cfDNA will be likely more useful in conjunction with CTCs to assess prognosis rather than each biomarker evaluated individually. In addition, the NGS of cfDNA is still too costly to be translated into routine clinical practice. In contrast, the absolute quantification by qPCR seems to be a more realistic and less time consuming method by which to detect blood-based indicators of prognosis. The same is true for the CTC methodology used in our study that allows CTC determination in a short period of time after blood withdrawal and does not require expensive equipment.

In conclusion, to the best of our knowledge, this is the first study to concomitantly investigate the CTCs and cfDNA in previously treated metastatic NSCLC receiving nivolumab. Our findings support the role of circulating biomarkers as helpful prognostic indicators in this setting. Moreover, we identified a subgroup of patients at low-risk of death, some of who were performing clinically well beyond PD, who particularly benefited from nivolumab treatment. Further large-scaled studies to confirm the clinical value of CTCs and cfDNA combinations are warranted.

## Figures and Tables

**Figure 1 jcm-08-01011-f001:**
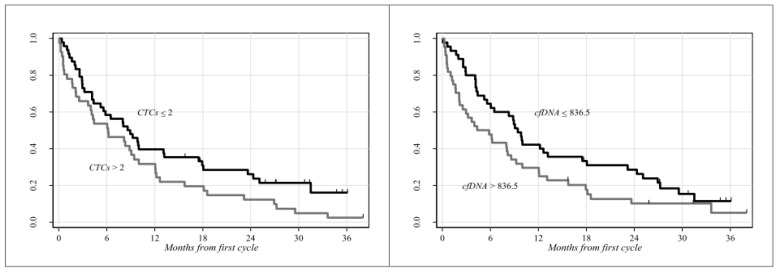
Clinical outcome estimated through the Kaplan-Meier analysis according to CTCs and cfDNA. Patients were categorized into two groups by means of the respective median baseline values. (**Left**): median survival times were 8.8 and 6.2 months for patients with ≤2 and >2 CTCs/3 mL of blood, respectively (*p* = 0.05). (**Right**): median survival times were 9.4 and 5.1 months for patients with ≤836.5 and >836.5 ng cfDNA/3 mL of plasma, respectively (*p =* 0.04).

**Figure 2 jcm-08-01011-f002:**
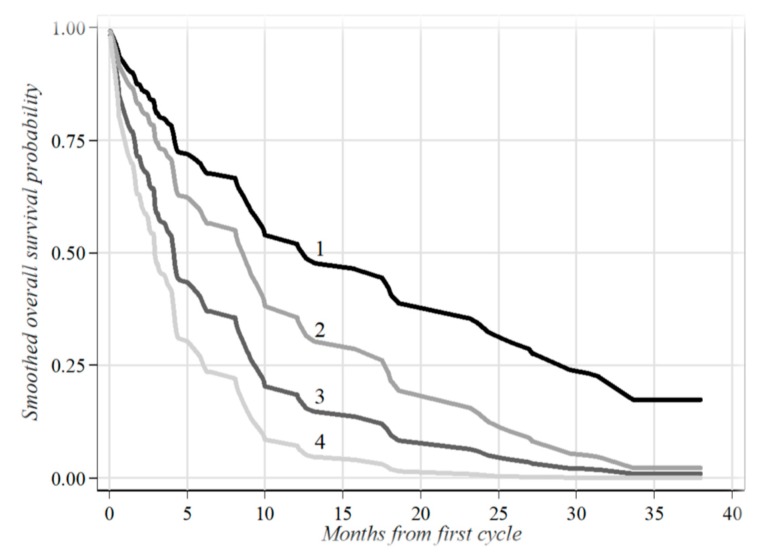
Survival probabilities estimated through the Cox regression modeling in four risk subgroups. The risk subgroups were identified by cross-classifying patients according to the two categories of both biomarkers evaluated at baseline (CTCs: ≤ 2 and > 2; cfDNA: ≤ 836.5 and > 836.5) considered as time-dependent covariates (*p* < 0.001). **1**—CTCs ≤ 2 and cfDNA ≤ 863.5, HR = 1.00 (Reference category): 27 patients; **2**—CTCs > 2 and cfDNA ≤ 863.5, HR = 1.72 (95% CI = 0.98-3.02): 17 patients; **3**—CTCs ≤ 2 and cfDNA > 863.5, HR = 2.64 (95% CI = 1.44–4.83): 21 patients; **4**—CTCs > 2 and cfDNA > 863.5, HR = 4.53 (95% CI = 2.11–9.73): 24 patients; HR: hazard ratio; 95% CI: 95% confidence intervals for HR. *p*-value: probability level associated with the likelihood ratio test. Ref.: reference category.

**Table 1 jcm-08-01011-t001:** Patients’ characteristics at baseline.

Baseline Characteristics	N (%)
**Patients**	89
**Median age** (range)	67 (44–86)
**Gender**	
Male	63 (70.8)
Female	26 (29.2)
**Histotype**	
Adenocarcinoma	63 (70.8)
Squamous cell carcinoma	25 (28.0)
Missing	1 (1.2)
**ECOG PS**	
0	31 (34.8)
≥1	58 (65.2)
**Smoking status**	
Never	9 (10.1)
Former	46 (51.7)
Smoker	28 (31.5)
Missing	6 (6.7)
**Number of metastases**	
1–2	32 (35.9)
3–4	40 (44.9)
5–9	16 (18.0)
Missing	1 (1.1)
**Prior lines of treatment**	
1	40 (44.9)
>1	45 (50.6)
Missing	4 (4.5)

ECOG PS: Eastern Cooperative Oncology Group Performance Status.

**Table 2 jcm-08-01011-t002:** Separate and joint prognostic effect of repeated measurements over time of CTCs and cfDNA on OS estimated through the Cox regression modeling.

Time-Dependent Biomarkers	Separate Effect			Joint Effect		
	HR	95% CI	*p*-Value	HR	95% CI	*p*-Value
**CTCs**			0.022			0.056
≤2	1.00	(Ref.)		1.00	(Ref.)	
>2	1.96	1.12–3.42		1.72	0.98–3.02	
**cfDNA**			<0.001			0.002
≤836.5	1.00	(Ref.)		1.00	(Ref.)	
>836.5	2.89	1.58–5.29		2.64	1.44–4.83	

HR: hazard ratio adjusted for age at start of nivolumab, gender, histotype, ECOG PS, smoking status, number of metastases, prior lines of treatment. 95% CI: 95% confidence intervals for HR. *p*-value: probability level associated with the likelihood ratio test. Ref.: reference category.

**Table 3 jcm-08-01011-t003:** Association of baseline CTCs and cfDNA with best response to nivolumab.

Best Response								
Baseline Biomarkers	Partial Response-Stable Disease		Progressive Disease		Total	OR	95% CI	*p*-Value
	N	%	N	%	N			
**CTCs**								
≤2	16	41	23	59	39	1	(Ref.)	0.405
>2	14	52	13	48	27	0.62	0.20–1.92	
**cfDNA**								
≤836.5	14	38	23	62	37	1	(Ref.)	0.191
>836.5	16	55	13	45	29	0.48	0.16–1.45	

Association was estimated through a multinomial logistic regression modelling using the median values of 2 CTCs and 836.5 ng cfDNA as cutoff points. OR: odds ratio adjusted for age at start of nivolumab, gender, histotype, ECOG PS, smoking status, number of metastases, prior lines of treatment; 95% confidence intervals for OR. *p*-value: probability level associated with the likelihood ratio test. Ref.: reference category.

**Table 4 jcm-08-01011-t004:** Separate and joint effects of baseline CTCs and cfDNA on OS of progressing patients estimated through the Cox regression modeling.

Baseline Biomarkers	Separate Effect			Joint Effect		
	HR	95% CI	*p*-Value	HR	95% CI	*p*-Value
**CTCs**			0.160			0.144
≤2	1.00	(Ref.)		1.00	(Ref.)	
>2	1.90	0.79–4.58		1.98	0.80–4.91	
**cfDNA**			0.005			0.005
≤836.5	1.00	(Ref.)		1.00	(Ref.)	
>836.5	4.00	1.54–10.40		4.08	1.55–10.72	

HR: hazard ratio adjusted for age at start of nivolumab, gender, histotype, ECOG PS, smoking status, number of metastases, prior lines of treatment. 95% CI: 95% confidence intervals for HR. *p*-value: probability level associated with the likelihood ratio test. Ref.: reference category.

**Table 5 jcm-08-01011-t005:** Survival probabilities estimated through the Cox regression modeling in four risk groups of progressing patients.

Risk Groups	Baseline Biomarkers		Overall Survival		*p*-Value
			HR	95% CI	
	**cfDNA**	**CTCs**			0.007
**1**	≤836.5	≤2	1.00	(Ref.)	
**2**		>2	1.98	0.80–4.90	
**3**	>836.5	≤2	4.08	1.55–10.71	
**4**		>2	8.09	2.06–31.71	

The risk subgroups were identified by cross-classifying patients according to the two categories of both biomarkers (CTCs: ≤ 2 and > 2; cfDNA: ≤ 836.5 and > 836.5). **1**—low (Reference category; Ref.): 15 patients; **2**—low-intermediate: 8 patients; **3**—high-intermediate: 8 patients; **4**—high: 5 patients; HR: hazard ratio; 95% CI: 95% confidence intervals for HR. *p*-value: probability level associated with the likelihood ratio test.

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
