# Peer review of "Prognostic Relevance of Circulating Tumor Cells and Circulating Cell-Free DNA Association in Metastatic Non-Small Cell Lung Cancer Treated with Nivolumab"

_jcm, 2019, doi:10.3390/jcm8071011_

Round 1
Reviewer 1 Report
Alama et al., present in this work the prognostic significance of the measurement of two blood biomarkers (CTCs number and cfDNA measurement) in NSCLC patients receiving Nivolumab.
The study is well designed and authors have used a corred methodology. These type of studies are necessary to identify those patients who could benefit from ICIs treatments.
The bibliography provided is of high quality and very up to date.
However, I would like authors to clarify some issues in the text of manuscript:
- The number of participants included in the study is small. Authors categorize patients into four risk subgroups, but do not show how many patients were included in each subgroup. This is important given the small sample size, since some subgroup may be underrepresented, mainly in patients with Progression Disease.
- This limitation should be indicated in the Discussion section.
- On the other hand, I would like the authors to indicate if there is any correlation between the number of CTCs and the amount of cfDNA in the patients included in the study.
- In page 7, line 193, authors indicate "Moreover, by considering the risk subgroups…, a strong risk of death (HR= 8.09....) was found for patients in the high category (Table 5)". However, according to Table 5, this seems to be true for both group 4 and group 3.
- Results in progressing patients seem to indicate that the amount of cfDNA is more important than the CTCs number, related to OS. Could you comment / criticize about this?
Reviewer 2 Report
Alama et. al. reported herein the use of circulating tumor cells and cell-free DNA as biomarkers for metastatic non-small cell lung cancer patients treated with an immune checkpoint inhibitor, nivolumab. This study provides a rationale for subsequent large-scaled studies to further confirm the clinical relevance of such a detection method.
It is highly recommended that the authors expand the introduction section and cited the related previous works. The authors should cite a similar work by Passiglia et. al.: "Monitoring blood biomarkers to predict nivolumab effectiveness in NSCLC patients" and a similar work regarding chemotherapy by Tong et. al. : "Prognostic significance of circulating tumor cells in non-small cell lung cancer patients undergoing chemotherapy". In the introduction, some recent review papers should be cited, e.g. Kapeleris et. al. : "The Prognostic Role of Circulating Tumor Cells (CTCs) in Lung Cancer".
Reviewer 3 Report
In this study the authors have analyzed CTCs and cfDNA from 89 lung cancer patients receiving ICI. Blood has been collected before therapy and at first and second evaluation. They showed that CTCs and the cfDNA have a prognostic value in this group of patients. The role of CTCs and cfDNA in lung cancer patients has attracted much focus recently, and may be extremely valuable in prediction of treatment response.
Comments to the paper:
The amount of cfDNA was estimated using hTERT as a reference gene. It is recommended to use multiple reference genes in order to minimize biases in qPCR analyses. The quantification of cfDNA would have been more robust if the authors had usedmultiple reference genes.
Table 2 shows the prognostic effect of CTCs and cfDNA on OS with repeated measurements over time estimated through the cox regression modeling. It is not clear how the repeated measurement of the two biomarkers is used in the cox regression modeling. The authors should explain in more details how the repeated measurement is used in this model. Is both first and second evaluation included? Samples from 89 patients were collected at baseline, but only 66 and 37/36 patient samples were analyzed at first and second evaluation. How did this affect the results in the repeated measurements, how many patients were in the risk groups in the model at second evaluation? How many patients in each risk group? How many of the patients were regarded as high-risk patients?
Is figure 2 base on baseline values of CTCs and cfDNA?
